# Point-Of-Care Urine LAM Tests for Tuberculosis Diagnosis: A Status Update

**DOI:** 10.3390/jcm9010111

**Published:** 2019-12-31

**Authors:** Michelle A. Bulterys, Bradley Wagner, Maël Redard-Jacot, Anita Suresh, Nira R. Pollock, Emmanuel Moreau, Claudia M. Denkinger, Paul K. Drain, Tobias Broger

**Affiliations:** 1FIND, 1202 Geneva, Switzerland; 2International Clinical Research Center, University of Washington, Seattle, WA 98105, USA; 3Institute of Disease Modeling, Bellevue, WA 98005, USA; 4Department of Laboratory Medicine, Boston Children’s Hospital, Boston, MA 02115, USA; 5Division of Tropical Medicine, Center of Infectious Diseases, University of Heidelberg, 69120 Heidelberg, Germany

**Keywords:** Tuberculosis (TB), diagnostics, Lipoarabinomannan (LAM), urinary tests

## Abstract

Most diagnostic tests for tuberculosis (TB) rely on sputum samples, which are difficult to obtain and have low sensitivity in immunocompromised patients, patients with disseminated TB, and children, delaying treatment initiation. The World Health Organization (WHO) calls for the development of a rapid, biomarker-based, non-sputum test capable of detecting all forms of TB at the point-of-care to enable immediate treatment initiation. Lipoarabinomannan (LAM) is the only WHO-endorsed TB biomarker that can be detected in urine, an easily collected sample. This status update discusses the characteristics of LAM as a biomarker, describes the performance of first-generation urine LAM tests and reasons for slow uptake, and presents considerations for developing the next generation of more sensitive and impactful tests. Next-generation urine LAM tests have the potential to reach adult and pediatric patients regardless of HIV status or site of infection and facilitate global TB control. Implementation and scale-up of existing LAM tests and development of next-generation assays should be prioritized.

## 1. Introduction

Tuberculosis (TB) has infected one-quarter of the world’s population and is the leading infectious cause of mortality worldwide [1]. Ten million people develop active TB each year, one million of whom are children [1]. An estimated 36% of new TB cases remain undiagnosed or unreported, partly due to the major limitations of current diagnostic tools [1]. Most conventional diagnostic tests for microbiological confirmation rely on sputum samples, which can be difficult to obtain and have low diagnostic sensitivity in children, patients with extrapulmonary TB (EPTB), and people living with HIV (PLHIV) [2]. EPTB occurs in one fifth of all incident TB cases, and the majority (60%) of EPTB patients do not have traceable TB in the lungs and sputum [3]. PLHIV experience higher rates of EPTB [1].

In 2014, the World Health Organization (WHO) called for the development of a “rapid biomarker-based non-sputum test capable of detecting all forms of TB by identifying characteristic biomarkers” (Table 1) [4]. The target product profile (TPP) specified that the test must be able to diagnose active TB and have a high specificity to allow initiation of treatment at the same clinical encounter or on the same day [4]. A point of care (POC) test that readily detects active TB would reduce diagnostic delays, interrupt transmission with appropriate therapy, and address many of the current gaps in global TB control.

## 2. Lipoarabinomannan in Active TB Disease 

Active *Mycobacterium tuberculosis (Mtb*) infection begins when *Mtb* enters the lungs via inhalation and invades the lung interstitial tissue where the process of infection evolves (Figure 1). This leads to the recruitment of an increasing number of immune cells to the lung parenchyma to form a granuloma (Figure 1). On the one hand, macrophages in the granuloma are capable of killing or at least controlling the growth of *Mtb* with the potential to ‘ward off’ infection from the rest of the body. On the other hand, granulomas are a growing collection of phagocytic cells that *Mtb* can infect and replicate within [5]. If the bacterial load becomes too great, the granuloma will fail to contain the infection, allowing *Mtb* to enter the bloodstream or the lymphatic system, disseminate to other extrapulmonary sites (Figure 1), or re-enter the respiratory tract to be released. The patient is now infectious and is said to have active TB disease. 

*Mtb* has a unique cell wall with multiple lipid-based molecules that create a thick ‘waxy’ surface [6]. A major component of this cell envelope is lipoarabinomannan (LAM), which represents up to 15% of the bacterial mass [7]. LAM is firmly but non-covalently attached to the inner membrane and extends to the exterior of the cell wall (Figure 1) [8] where it interacts as a potent virulence factor that modulates the host immune response and plays an important role in the pathogenesis of *Mtb* infection [9]. The exact molecular structure and size of LAM in vivo is unknown and might differ in different parts of the body. If produced in vitro, LAM’s average molecular weight is 17.4 kilodaltons, but the molecule is heterogeneous in size, branching pattern, acylation, and phosphorylation on the arabinan and mannan portions [10,11]. LAM has four structural domains (Figure 1): (I) the glycophospholipid anchor, which attaches the molecule non-covalently to the inner membrane, (II) the attached mannan core, which is highly conserved across mycobacterial species, and (III) the variable branching arabinan side chains with (IV) variable capping motifs that give rise to the intra- and inter-species diversity of LAM molecules [12]. According to the capping motifs, LAM can be classified into three structural families: LAM from fast-growing, non-pathogenic species, such as *M. smegmatis*, have uncapped ends (AraLAM) or inositol phosphate caps (phosphatidyl-myo-inositol capped LAM, or PILAM) whereas LAM from slow-growing mycobacteria (*Mtb, M. leprae, M. avium*, and *M. kansasii*) is modified with one to three α (1→2)-linked mannopyranose (Man*p*) units (ManLAM). *Mtb* ManLAM can contain an additional cap modification, 5-deoxy-5-methylthio-xylofuranose (MTX), attached to the terminal Man*p* [13,14,15].

As replicating *Mtb* degrades, LAM circulating in the blood is filtered across the glomerular basement membrane of the kidneys into urine (Figure 1). The presence of LAM in urine can also be a result of renal *Mtb* infection, as has been shown in autopsy studies [16]. 

There are few studies reporting LAM concentrations in clinical specimens and direct comparisons between different sample types and assays are complicated by the absence of standardized LAM control materials, sample panels, and reference assays. Four recent studies used the same purified LAM material for calibration and similar antibody reagents for immunoassay-based LAM detection (though different detection platforms) and reported LAM concentrations in sputum [33], blood [34,35], and urine [36] in subjects with active pulmonary TB, allowing for a rough comparison of LAM concentration ranges. For sputum, Kawasaki and colleagues showed that an immunoassay with a cut-off of 15 pg/mL detected all smear-positive and 50% of smear-negative TB patients [33] and sputum LAM concentrations ranged from 15.4 pg/mL to 1,869,000 pg/mL (median 5512 pg/mL). LAM concentration in sputum was linearly correlated to colony forming units (CFU) with 1 pg/mL of LAM correlating to 8 CFU/mL, suggesting that one *Mtb* bacterium contains approximately 125 fg of LAM [33]. Brock et al. [35] developed a sensitive serum assay using a Single Molecule Array (SIMOA); using a cutoff of 2.3 pg/mL, the assay sensitivity in smear-positive TB patients was 40%, and 60% in HIV-positive/smear-positive TB patients at 100% specificity. Serum LAM concentrations in this study were highest in HIV-positive/smear-positive patients (median 3.97 pg/mL). Broger and colleagues [34,36] developed sensitive immunoassays and compared serum LAM to urine LAM concentrations in matched samples from smear-positive TB patients. The serum assay detected LAM in 55% of smear-positive patients with concentrations of 6 pg/mL to 70,000 pg/mL but 45% of patients were below the cut-off of 6 pg/mL, suggesting a median concentration in serum of roughly 10 pg/mL. In the same sample set, the urinary LAM assay with a cut-off of 11 pg/mL showed that nearly all patients (93%) had concentrations in the range of 12 pg/mL to 90,000 pg/mL (median 111 pg/mL). In summary, data suggest roughly 50-times higher median LAM concentrations in sputum compared to urine, roughly 500-times higher median LAM concentrations in sputum compared to serum, and roughly 10-times higher median LAM concentrations in urine compared to serum. However, further confirmation is required as detection might have been influenced by reagents (e.g., cross-reactivity of antibodies with LAM from non-*Mtb* bacteria), structural differences of LAM in different sample types, assay designs, and detection platforms.

LAM concentrations in the different body fluids can be drastically influenced by factors such as bacterial burden, infection site (i.e., pulmonary vs. extrapulmonary TB, including urinary tract disease), and co-morbidities such as HIV. There is evidence that HIV/TB co-infected patients with immunosuppression and disseminated TB have higher LAM concentrations in their urine [37]. Biological mechanisms to explain this are not fully understood; whether this is due to an overall higher bacterial body burden or renal TB infection is unclear [16,38]. While HIV co-infection or urinary tract disease are not necessary for LAM antigenuria, they both can lead to higher urine LAM concentrations.

## 3. First-Generation LAM Tests

Urine LAM has been evaluated as a diagnostic biomarker for TB testing since 1997 [39]. In 2003, Chemogen Inc. (South Portland, OR, USA) developed a lab-based urine ELISA for LAM detection, which was later commercialized by Inverness Inc. as the Clearview^®^ TB ELISA assay. In 2010, Inverness changed its name to Alere and developed the first rapid POC LAM test, the Determine TB LAM Ag lateral flow assay (“AlereLAM”). In 2017, Alere was acquired by Abbott Diagnostics (Lake Bluff, USA). The AlereLAM test likely uses the same polyclonal antibodies as the Clearview ELISA, and there is high observed agreement between the two assays [40]. The AlereLAM test meets the operational characteristics of the WHO TPP for a biomarker-based, non-sputum TB test, but falls short on test accuracy (Table 1). In 2019, a meta-analysis by Bjerrum et al. [21] compared pooled sensitivity and specificity against a microbiological reference standard (MRS), defined as a positive TB culture or TB nucleic acid amplification test (NAAT). When compared against an MRS, pooled sensitivity and specificity in HIV-positive adults were 42% (95% CI: 31–55%) and 91% (95% CI: 85–95%), respectively [21]. It is likely that the MRS from Bjerrum and colleagues’ meta-analysis underestimates AlereLAM specificity; an earlier meta-analysis reported 96–98% specificity against a composite reference standard (CRS) [26].

Among symptomatic participants with CD4 counts >200 cells/µL, pooled sensitivity and specificity were 16% (95% CI: 8–31%) and 94% (95% CI: 81–97%) [21]. Among symptomatic participants with CD4 counts ≤200 cells/µL, pooled sensitivity and specificity were 45% (95% CI: 31–61%) and 89% (95% CI: 77–94%) [21]. Among children, data were limited, but five prospective pediatric cohort studies found a pooled sensitivity of 47% (95% CI: 27–69%) and specificity of 82% (95% CI: 71–89%) [24]. A possible reason for the lower specificity in children could be cross-reactivity of AlereLAM with bacteria from perineal skin or stool that contaminate the urine sample during collection, as urine bags may remain on the skin for several hours until the child produces urine [41], or limitations of the reference standard. Based on these meta-analysis results [21], the WHO announced updated guidelines for use of AlereLAM in 2019, which are summarized in Table 2 [42]. 

To date, three randomized controlled trials (RCT) of test implementation have assessed the effect of urine AlereLAM testing on mortality outcomes. Two studies randomly assigned patients 1:1 to either the standard of care (SOC) arm (a combination of Xpert-MTB/RIF, smear microscopy, and culture in the first RCT, and Xpert-MTB/RIF alone in the second RCT) or the interventional arm (SOC plus LAM testing). The first RCT (LAMRCT, [28]) enrolled HIV-positive adults with at least one TB symptom in ten hospitals across South Africa, Tanzania, Zambia, and Zimbabwe. Peter et al. [28] found overall 8-week mortality occurred in 25% of the SOC arm and 21% in the LAM group, and concluded that “LAM-guided initiation of anti-TB treatment” reduced absolute risk of mortality by 4% (95% CI: 1–7%) and relative risk by 17% (95% CI: 4–28%). The second RCT (STAMP trial, [29]) enrolled HIV-positive adults from two hospitals in Malawi and South Africa and found overall 8-week mortality occurred in 21% of the SOC arm and 18% in the LAM group; an adjusted risk reduction of −2.8% (95% CI: −5.8–0.3%; *p* = 0.074) [29]. This study found that urine LAM testing significantly reduced the risk of mortality in three pre-specified clinical subgroups: patients with CD4 counts <100 cells/µL, by −7.1% (95% CI: −13.7% to −0.4%; *p* = 0.036), severely anemic patients, by −9.0% (95% CI: −16.6% to −1.3%; *p* = 0.021), and those with clinically suspected TB, by −5.7% (95% CI: −10.9% to −0.5%; *p* = 0.033) [29]. The authors concluded that urine LAM could reduce mortality in these high-risk groups and argued for a use for screening in all HIV-positive inpatients. A third RCT (TB Fast Track, [30]) randomly assigned HIV-positive adults (aged ≥18 years) with CD4 counts of ≤150 cells/μL, who had not had antiretroviral therapy (ART) in the past 6 months 1:1 to either routine care or the interventional arm [30]. In the interventional arm, nurses assessed participants on the basis of tuberculosis symptoms, body-mass index, point-of-care hemoglobin concentrations, and urine AlereLAM results. Participants in the interventional arm classified as having high probability of tuberculosis were recommended to start tuberculosis treatment immediately, followed by ART two weeks later; participants classified as medium probability were recommended to have symptom-guided investigation; and participants classified as low probability were recommended to start ART immediately. Although the intervention substantially increased coverage of tuberculosis treatment in this high-risk population, it did not reduce mortality, and the authors proposed to prioritize the development of more sensitive TB diagnostic tests suitable for use by nurses in primary health-care clinics.

Despite the initial WHO recommendation in 2015 [24] and the availability of Global Fund and The President’s Emergency Plan for AIDS Relief (PEPFAR) funding for test procurement, the rollout and uptake of the AlereLAM test has been slow [43]. Only approximately 400,000 AlereLAM tests were sold in 2018 [44]. The majority of uptake has come from three high-burden countries: South Africa, Uganda, and Kenya. According to the Treatment Action Group’s (TAG’s) March 2019 report [17], an in-depth review of the Global Fund’s grant documents included mention of LAM procurement in six additional countries (Burundi, Cameroon, Eswatini, Guatemala, Ukraine, and Vietnam), and PEPFAR country operation plans included mention of LAM procurement in six other countries (Côte d’Ivoire, Democratic Republic of the Congo, Eswatini, Kenya, Malawi, and Zambia). The lack of widespread POC LAM implementation is related to multiple factors, including the conditional WHO recommendation and the lack of coordination between National TB Programs (NTPs) and HIV programs in low and middle-income countries [45]. Some NTP representatives consider LAM to be a “niche test” with clinical utility limited to a small population. LAM testing has increased as a result of the 2017 WHO HIV Advanced Care guidance [46] inclusion of AlereLAM test usage within its algorithms. The uptake has been further boosted by the involvement of patient advocacy groups such as TAG and others [17,47]. The addition of AlereLAM to the WHO Essential Diagnostic List (EDL) [48] and to the Global Drug Facility TB Diagnostics Catalog in August 2018 may further boost procurement [49]. These developments, together with the recently extended WHO recommendation for 2019 (Table 2, [42]), and the consequent policy and stakeholder support globally, provide support for both increased use of AlereLAM and development of a better-performing LAM test in the near future [50].

## 4. The Need for Next-Generation, Highly Sensitive, and Specific LAM Tests

The fundamental goal of next-generation LAM tests is to achieve a high diagnostic accuracy (sensitivity and specificity) while maintaining the operational characteristics of an easy-to-use, rapid, biomarker-based, non-sputum POC test in order to fulfill the WHO high priority TPP requirements (Table 1).

Several recent research reports indicate that lower detection limits will translate into higher diagnostic sensitivity and demonstrate necessary measures to improve the analytical sensitivity. First, Paris et al. developed a sample preparation device that concentrates antigens and detects LAM down to 14 pg/mL, resulting in 95% sensitivity at 80% specificity in a study of 48 HIV-negative TB-positive subjects and 58 TB-negatives with other respiratory diseases and healthy controls [51]. Shapiro et al. developed a device to concentrate and detect LAM in urine and reported 52% sensitivity in HIV-negative TB-positive persons at 67% specificity in a study with 93 patients [52]. Connelly et al. developed a prototype POC urine assay consisting of a concentration step and lateral flow assay and reported 60% sensitivity at 80% specificity in a cross-sectional study with 86 HIV-positive and 206 HIV-negative patients [53]. Hamasur et al. used magnetic particles to remove undisclosed ‘urine inhibitors’ in combination with LAM enrichment methods and reached 65.5% sensitivity at 84% specificity in a study of 45 HIV-positive and 74 HIV-negative patients [54]. While all these assays showed promising sensitivities (65–95%) they all have specificities below 85% and would not meet the minimal target of ≥98% of the biomarker TPP (Table 1).

In contrast, Sigal et al. developed a sensitive research assay that approaches the performance targets of the biomarker TPP. The assay targets the MTX-LAM epitope (Figure 1) in urine and reached 93% sensitivity and 97% specificity in a case-control study with 40 TB-positive and 35 HIV-negative patients with a cut-off of 11 pg/mL [36]. The study underlines the importance of using reagents that target *Mtb*-characteristic LAM epitopes to maintain a high diagnostic specificity at very low assay cut-offs. 

This work by Sigal et al. informed the development of the Fujifilm SILVAMP TB LAM test (“FujiLAM”; Fujifilm, Tokyo, Japan, Figure 1), which is a well-advanced next-generation LAM test (CE-marked) [20,27,55] (Table 1). The assay combines a pair of high-affinity monoclonal antibodies directed towards the largely *Mtb*-specific MTX-LAM epitopes and a silver-based amplification step that increases the visibility of test and control lines of a lateral-flow assay to reach a cut-off around 30 pg/mL [27]. This enables the detection of approximately 30-fold lower concentrations of LAM in urine compared to the AlereLAM test. FujiLAM showed approximately 30% improved diagnostic sensitivity in HIV-positive inpatients compared to AlereLAM at 95.7% specificity [27]. Additional studies in HIV-infected outpatients [31] and a recent meta-analysis of 1595 HIV-positive inpatients and outpatients [22] have confirmed FujiLAM’s superiority over AlereLAM. The meta-analysis reported 70.7% (95% CI: 59.0–80.8%) sensitivity for FujiLAM, a sensitivity increase of 35.8% compared to AlereLAM [22]. FujiLAM also showed high performance for the detection of EPTB in HIV-infected patients [56]. Two recent studies [31,32] showed that FujiLAM could have rapidly diagnosed TB in up to 90% of HIV-positive patients that died within 3 to 6 months, whereas the probability of survival in the case of a FujiLAM-negative result was 86–97%. A limitation of published FujiLAM studies is their use of biobanked samples and, accordingly, testing in a laboratory setting. Notably, a recent study with a cohort of 182 patients from Zambia demonstrated that the use of biobanked specimens delivers nearly equivalent results compared with fresh samples [57]. Prospective studies now need to confirm the initial findings, quantify FujiLAM’s effect on clinical outcomes, establish performance in HIV-negative patients and children, and assess the feasibility for POC implementation in a variety of clinical settings. Furthermore, cost-effectiveness analyses and impact modeling are needed. WHO evaluation of FujiLAM is expected in Q4/2020 and market entry in Q1/2021 [20]. Interestingly, Abbott, the manufacturer of AlereLAM, announced the development of a next-generation LAM test to extend the indication to the HIV-negative population but have not communicated a launch date [20].

## 5. Considerations for Evaluating LAM Assays

There are several common challenges and limitations faced in diagnostic accuracy studies of LAM tests. One major difficulty in evaluating non-sputum-based tests is an imperfect reference standard, particularly for patients with EPTB and paucibacillary disease including PLHIV and children. Studies assessing AlereLAM and other non-sputum-based tests have discussed this limitation being responsible for at least some of the false-positive results as defined by the index test [58]. In order to improve upon this, a combination of a microbiological reference standard (MRS) and a composite reference standard (CRS) or latent class analysis should be considered [59,60]. The MRS should combine results from several pulmonary samples (at least two, and including sputum induction if needed) as well as extrapulmonary samples (such as urine, blood, or others as indicated by the patient’s presentation) to perform multiple Xpert tests and mycobacterial cultures to define ‘Definite TB’. The CRS combines this with chest X-ray, clinical suspicion, and treatment initiation to define ‘Possible TB’ [60]. A patient follow-up period of 8 to 12 weeks or longer, including observation of clinical improvement in the absence of treatment, is critical to establish a ‘Non-TB’ diagnosis [60]. Cross-reactivity to other pathogens and colonizing organisms also needs to be critically evaluated in both analytical and clinical studies. Importantly, LAM assays can diagnose TB in patients unable to produce sputum, therefore, the analysis of diagnostic yield is important and patients unable to produce sputum should not be excluded as this group might benefit most from urinary LAM testing. Also, evaluation of incremental diagnostic yield of urinary LAM diagnostics in combination with sputum-based diagnostics should be considered. Other common considerations are user subjectivity in test interpretation if the readout is visual, as is the case for AlereLAM and FujiLAM, and ensuring generalizability across settings of implementation.

## 6. Future Directions and Potential Impact of LAM Tests

The impact of any large-scale rollout of a next-generation LAM test will ultimately be measured by reduction in disease mortality and incidence at the population level. In high-burden settings, this will depend significantly on achieving reductions in TB transmission through a combination of earlier case identification and reduction in loss-to-follow-up from the point of diagnosis to the initiation of treatment. A POC diagnostic test could significantly reduce initial loss-to-follow-up in the care cascade. Analyses of country-level data in South Africa have shown losses of approximately 12% between diagnosis and treatment initiation, which could potentially be diminished by POC testing [61,62]. Importantly, the ability to reduce these losses will also depend on other aspects of the overall care cascade such as the ability to perform drug susceptibility testing in LAM-positive patients. Modelling of intensified case-finding among HIV clinic attendees in South Africa has suggested reductions in mortality of up to 30% over 10 years if LAM test sensitivity is near the biomarker TPP of 65% [63]. Analyses of the potential impact of introducing LAM into the diagnostic and care cascades will need to view LAM testing in the context of its placement in specific health care systems and its relationship to other available diagnostics.

## 7. Conclusions

Urine LAM is a promising TB diagnostic biomarker for use in a POC TB test, with the potential to reach adult and pediatric patients regardless of HIV status or site of infection. The ultimate goal of a urine LAM test is to achieve sensitivity high enough to reach and benefit all TB patients while remaining specific, fast, simple, and affordable to use. FujiLAM offers greater sensitivity than AlereLAM and is therefore likely to extend the indication of TB LAM testing beyond seriously ill HIV-positive patients. However, several research questions remain to be addressed (Table 3). With dedicated research driving the next generation of urine LAM diagnostics for clinic-based TB detection, there is a clear path forward to making a significant public health impact and facilitating global TB control. Implementation and scale-up of existing LAM tests and development of next-generation assays should be prioritized.

## Figures and Tables

**Figure 1 jcm-09-00111-f001:**
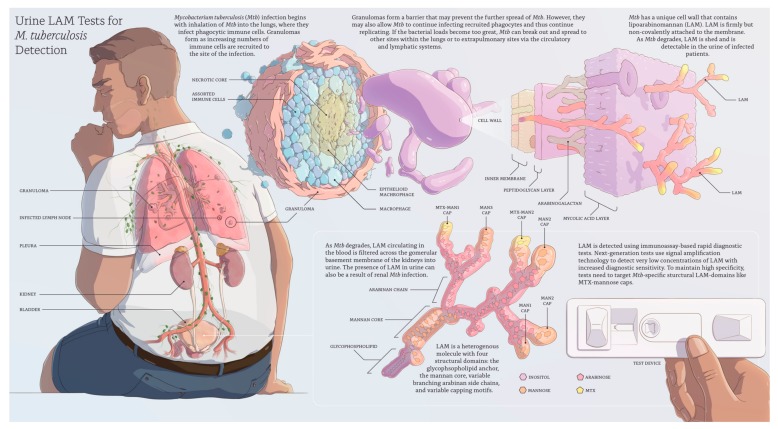
Overview of lipoarabinomannan (LAM) detection in urine for the diagnosis of active tuberculosis (developed by Digizyme, Inc., Boston, MA, USA and FIND, Geneva, Switzerland).

**Table 1 jcm-09-00111-t001:** Status of the available AlereLAM, FujiLAM, and sputum-based tuberculosis (TB) diagnosis vs. biomarker target product profile (TPP) [4]. WHO: World Health Organization; PLHIV: people living with HIV; TPP: Target product profile; CRS: Composite reference standard; IVD: in vitro diagnostic; “✓” indicates that the TPP minimal criteria is met; “*x*” indicates that the TPP minimal criteria is not met; “?” indicates that it is currently unclear whether the TPP criteria is met.

	WHO Target Product Profiles (TPPs) Minimal Criteria: “Biomarker Test”	Abbott Determine TB LAM Ag (AlereLAM)		Fujifilm SILVAMP TB LAM Assay (FujiLAM)		Sputum-Based TB Diagnosis (Smear Microscopy; Xpert MTB/RIF; Xpert MTB/RIF Ultra)	
Price per test (ex-works)	<US$6.00	✓	US$3.00–3.50 [17] (likely cost-effective among hospitalized HIV-postive patients)	?	Fujifilm has not yet released information on the price (initial cost-effectiveness modelling data presented [18])	*x*	Xpert MTB/RIF US$9.98 [19]
Regulatory requirements and availability	Registered for in vitro diagnostic (IVD) use	✓	CE-IVD marked IVD WHO recommendation (Table 2) On the market	?	CE-IVD marked IVD WHO evaluation expected Q4/2020 Market entry expected Q1/2021 [20]	✓	CE-IVD marked IVD Several WHO recommendations On the market
Equipment	Ideally instrument free and small, portable, or handheld, <1 kg, <US$500 instrument acceptable	✓	Instrument free	✓	Instrument free	*x*	US$17,500.00 [19] (GeneXpert platform plus laptop)
Sensitivity in HIV positive patients (independent of CD4 count)	≥65% (not specifically defined for HIV-positives in the TPP)	*x*	42% [21]	✓	70.7% [22]	✓	90% (Xpert MTB/RIF Ultra) [23] 77% (Xpert MTB/RIF) [23] 47% (Microscopy) [24]
Sensitivity in HIV negative patients	≥65%	*x*	18% [25]	?	No information published. Studies ongoing.	✓	91% (Xpert MTB/RIF Ultra) [23] 90% (Xpert MTB/RIF) [23]
Specificity	≥98%	?	96–98% against CRS [26] (likely meeting the target as specificity might be underestimated due to limitations of the reference standard)	?	95.7% against CRS [27] (likely meeting the target as specificity might be underestimated due to limitations of the reference standard)	✓	96% (Xpert MTB/RIF Ultra) [23] 98% (Xpert MTB/RIF) [23] 98% (Microscopy) [24]
Day 1 diagnostic yield in HIV-positive inpatients (TB patients diagnosed on the first day they present)	No target defined in TPP		43.3% [27]		64.5% [27]		26.2% (Xpert MTB/RIF) [27] 19.1% (Microscopy) [27]
Outcome/ / mortality impact	No target defined in TPP		Mortality impact shown in hospitalized PLHIV but not in more general populations. A positive result is associated with increased risk of mortality [28,29,30]		No impact studies available. A positive result is associated with increased risk of mortality [31,32]		Unclear
Sample type	Non-sputum	✓	Urine	✓	Urine	*x*	Sputum
Time-to-result	<60 min	✓	25 min	✓	50–60 min	*x*	100 min (Xpert MTB/RIF and Xpert MTB/RIF Ultra)
Number of steps	Limited number of steps, no precise measuring	✓	2 steps	✓	5 steps	*x*	Xpert MTB/RIF and Xpert MTB/RIF Ultra: 11 steps Microscopy: >10 steps
Setting and infrastructure needs	Primary health-care clinics with lab or microscopy center or higher levels	✓	Simple to use lateral flow assay	✓	Simple to use lateral flow assay	*x*	Laboratory required Electricity required Equipment susceptible to dust and shock

**Table 2 jcm-09-00111-t002:** Updated 2019 WHO guidelines for the use of AlereLAM [42].

**In inpatient settings**, WHO strongly recommends using lateral flow urine lipoarabinomannan assay (LF-LAM) to assist in the diagnosis of active TB in HIV-positive adults, adolescents, and children: with signs and symptoms of TB (pulmonary and/or extrapulmonary) (**strong** recommendation; moderate certainty in the evidence about the intervention effects); or with advanced HIV disease or who are seriously ill (**strong** recommendation; moderate certainty in the evidence about the intervention effects); or irrespective of signs and symptoms of TB and with a CD4 count <200 cells/mm^3^ (**strong** recommendation; moderate certainty in the evidence about the intervention effects).
**In outpatient settings, ** WHO suggests using LF-LAM to assist in the diagnosis of active TB in HIV-positive adults, adolescents, and children: with signs and symptoms of TB (pulmonary and/or extrapulmonary) or “seriously ill” (**conditional** recommendation; low certainty in the evidence about test accuracy); and irrespective of signs and symptoms of TB and with a CD4 count of less than 100 cells/mm^3^ (**conditional** recommendation; very low certainty in the evidence about test accuracy).
**In outpatient settings**, WHO recommends **against** using LF-LAM to assist in the diagnosis of active TB in HIV-positive adults, adolescents, and children: without assessing TB symptoms (**strong ** recommendation; very low certainty in the evidence about test accuracy); and without TB symptoms and unknown CD4 cell count or without TB symptoms and CD4 count ≥200 cells/mm^3^ (**strong ** recommendation; very low certainty in the evidence about test accuracy); and without TB symptoms and with a CD4 count of 100–200 cells/mm^3^ (**conditional ** recommendation; very low certainty in the evidence about test accuracy).

**Table 3 jcm-09-00111-t003:** Research, development, and implementation questions for future urine Lipoarabinomannan (LAM) Tests. POC: point of care.

**Research and development questions**	
1	What concentrations of LAM need to be detected to meet TPP sensitivity in HIV-negative populations?
2	Can sensitive, specific, low-cost, simple, and rapid platform alternatives be developed that substantially improve POC detection compared to conventional lateral-flow assays?
3	Which of the currently available antibodies yield the best performance in immunoassays?
4	What simple and POC-amenable specimen processing steps would improve the availability for detection or increase the concentration of LAM in clinical samples?
5	Are multiple molecular species of LAM present in clinical specimens, implying the need for polyclonal antibodies or multiple sets of monoclonal antibodies?
6	What purified LAM antigen preparations best mimic what is found in patient samples?
7	What is the molecular structure of LAM released from *M. tuberculosis* in vivo?
**Implementation and public health impact questions**	
	**Patient-related questions**
8	What is the target population and distribution? Which populations will benefit from this tool? Which populations could eventually benefit?
9	How much more impact will a next-generation test have on mortality risk reduction?
10	Can LAM tests be used to monitor treatment adherence and/or completion?
11	What are the diagnostic yields of LAM tests alone and in combination with smear microscopy or Xpert?
	**Operational and health systems questions**
13	What is the intended level of healthcare facility and user level of training required?
14	How much time is saved by using this tool versus other standard-of-care tools?
	**Policy and access questions**
15	What global institutions, technical experts, and financing organizations are considered key influencers in the global market for this product? How are we engaging with them?
16	What are the early-adopter countries that may drive product uptake and expansion? Which countries are potential new markets for next-generation LAM tests?
17	How will next-generation LAM tests integrate into current WHO guidelines and TPPs for TB detection?

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
