# Peer review of "Point-Of-Care Urine LAM Tests for Tuberculosis Diagnosis: A Status Update"

_jcm, 2019, doi:10.3390/jcm9010111_

Round 1

Reviewer 1 Report

This article is an extremely well-documented mini-review on the urine LAM tests developed for point-of-care diagnosis of tuberculosis. The scientific content is sound, the authors perform a very objective survey and analysis of literature data, the references are appropriate. The article represents a very good source of information for every reader interested in this particular field. Criticisms are few and this mini-review is acceptable after minor modifications.

The following minor points should be addressed:

l.58: read outer membrane l.64 and Figure 1: read glycosyl-phosphatidyl-myo-inositol anchor instead of glycophospholipid anchor (glycophsopholipid in Figure 1) l.64: attaches instead of binds l.67-71: LAM from fast-growing, non-pathogenic species, have uncapped ends (AraLAM), such as M. chelonae, or phospho-myo-inositol caps (PILAM) such M. smegmatis; see PMID:12765785; 25629008 During the reviewing process, a paper was published by Dr Chatterjee and colleagues on the “Comparative Structural Study of Terminal Ends of Lipoarabinomannan from Mice Infected Lung Tissues and Urine of a Tuberculosis Positive Patient.” PMID: 31762254. The authors may cite this new article.

Author Response

This article is an extremely well-documented mini-review on the urine LAM tests developed for point-of-care diagnosis of tuberculosis. The scientific content is sound, the authors perform a very objective survey and analysis of literature data, the references are appropriate. The article represents a very good source of information for every reader interested in this particular field. Criticisms are few and this mini-review is acceptable after minor modifications.

The following minor points should be addressed:

Author Response: Thank you for the interest and insightful feedback.

l.58: read outer membrane l.64 and Figure 1: read glycosyl-phosphatidyl-myo-inositol anchor instead of glycophospholipid anchor (glycophsopholipid in Figure 1)

Author Response: We have updated Figure 1 and terminology accordingly. The previous label of “outer membrane” was incorrect and confusing; this has now been corrected to “mycolic acid layer”. While consulting with experts, we also noted some minor inconsistencies with regards to the placement of LAM in the cell wall, and we believe that Figure 1 is now accurate and fully up to date. 

l.64: attaches instead of binds

Author Response: The terminology has been updated from “binds” to “attaches”.

l.67-71: LAM from fast-growing, non-pathogenic species, have uncapped ends (AraLAM), such as M. chelonae, or phospho-myo-inositol caps (PILAM) such M. smegmatis; see PMID:12765785; 25629008 During the reviewing process, a paper was published by Dr Chatterjee and colleagues on the “Comparative Structural Study of Terminal Ends of Lipoarabinomannan from Mice Infected Lung Tissues and Urine of a Tuberculosis Positive Patient.” PMID: 31762254. The authors may cite this new article.

Author Response: Thank you for this comment, we have updated the text and refer to the suggested paper by Dr. Chatterjee et al. (“Comparative Structural Study of Terminal Ends of Lipoarabinomannan from Mice Infected Lung Tissues and Urine of a Tuberculosis Positive Patient.” PMID: 31762254).

Reviewer 2 Report

General comments:

In this review manuscript, the authors discussed the LAM test for TB diagnosis. Although the rationale for the current paper is essential, however, the manuscript has some minor issues that need to be addressed before it gets to publish. I hope the comments will help the author to make it more clear.

Specific comments:

Abstract:

The title is little dry to me. I am not convinced about the word “mini-review”, perhaps, a narrative review would be better. What did the review find? It is not clear from the abstract about the results. Can you please expand it and talk about your result?

Introduction:

Page 2 of 13, lines 48-51, seems almost the same information. Please revise or clarify. Do you have any specific keywords and guideline that used to search the articles? It should be worth to include a search strategy or flow chart to give a general overview. This is also important to show that the author did not missed any recent articles.    I disagree with the statement from lines 79-81, page 2 of 13. Please justify your answer. Lines 107-109, page 3 of 13, is not clear, please revise.   What're new and novel findings here? I can see some limitations, but the author even didn’t mention. This should be in the discussion. What is the strength of this study? Reference is inconsistent and needs to correct according to MDPI requirement.

Author Response

REVIEWER #2

In this review manuscript, the authors discussed the LAM test for TB diagnosis. Although the rationale for the current paper is essential, however, the manuscript has some minor issues that need to be addressed before it gets to publish. I hope the comments will help the author to make it more clear.

Author Response: Thank you very much for the reviewer’s comments, which have helped to improve the manuscript.

Specific comments:
Abstract:
The title is little dry to me. I am not convinced about the word “mini-review”, perhaps, a narrative review would be better. What did the review find? It is not clear from the abstract about the results. Can you please expand it and talk about your result?

Author Response: We have changed the title to “Point-of-Care Urine LAM Tests for Tuberculosis Diagnosis: A Status Update” and specified this in the abstract. We were invited to write and submit a brief summary on the use of urine LAM as a biomarker for TB detection; a systematic review following PRISMA guidelines would have been out of scope. We did ensure that we did not miss relevant articles by testing search terms (see below). In the abstract, we specify that this status update “discusses the characteristics of LAM as a biomarker, describes the performance of first-generation urine LAM tests and reasons for slow uptake, and presents considerations for developing the next generation of more sensitive and impactful tests.” This paper does not have results to discuss.

Introduction:
Page 2 of 13, lines 48-51, seems almost the same information. Please revise or clarify.

Author Response: The first sentence describes host defense in granuloma. The second sentence describes Mtb proliferation in granuloma. Describing the difference between defense and proliferation in granuloma is important and we kept as is.

Do you have any specific keywords and guideline that used to search the articles? It should be worth to include a search strategy or flow chart to give a general overview. This is also important to show that the author did not missed any recent articles.    

Author Response: To keep the paper short and focused, we did not conduct a formal systematic review. Thanks to Reviewer 2’s suggestion above, we have now clearly specified this is a “status update” rather than a “mini-review” in the title and abstract, as there was no systematic searching in the preparation of this article. To ensure that we were still capturing relevant and recent papers, we checked using the following search terms in Pubmed: “(tuberculosis or TB) AND (lipoarabinomannan or LAM) AND (test OR assay OR antigen OR Ag OR lateral flow assay* OR urine antigen OR point of care) AND (accuracy OR sensitivity OR specificity OR yield OR diagnos* OR screening) AND (HIV negative)”. This helped us ensure that we did not miss publications of relevance; and these search terms did not identify any additional papers.

I disagree with the statement from lines 79-81, page 2 of 13. Please justify your answer.

Author Response: We agree and have updated the section. It now reads, “There are few studies reporting LAM concentrations in clinical specimen and direct comparisons between different sample types and assays are complicated by the absence of standardized LAM control materials, sample panels and reference assays.”

Lines 107-109, page 3 of 13, is not clear, please revise.

Author Response: We agree and have updated the section. It now reads, “There is evidence that HIV/TB coinfected patients with immunosuppression and disseminated TB have higher LAM concentrations in their urine[21]. Biological mechanisms to explain this are not fully understood; whether this is due to an overall higher bacterial body burden or renal TB infection is unclear[16, 22].”

What're new and novel findings here? I can see some limitations, but the author even didn’t mention. This should be in the discussion. What is the strength of this study?

Author Response: As per Reviewer 2’s suggestion, we have changed the title to clarify that this is not a systematic review, and therefore does not have novel findings.

Reference is inconsistent and needs to correct according to MDPI requirement.

Author Response: We downloaded the MDPI Style into EndNote. All references should now have been corrected and consistent with MDPI requirements.

Round 2

Reviewer 2 Report

The author responses to the reviewer's comments and I do not have any further comments.